# Diagnostic Blood Biomarkers in Alzheimer’s Disease

**DOI:** 10.3390/biomedicines10010169

**Published:** 2022-01-13

**Authors:** Jung Eun Park, Tamil Iniyan Gunasekaran, Yeong Hee Cho, Seong-Min Choi, Min-Kyung Song, Soo Hyun Cho, Jahae Kim, Ho-Chun Song, Kyu Yeong Choi, Jang Jae Lee, Zee-Yong Park, Woo Keun Song, Han-Seong Jeong, Kun Ho Lee, Jung Sup Lee, Byeong C. Kim

**Affiliations:** 1Department of Biomedical Science, Chosun University, Gwangju 61452, Korea; jepark@chosun.ac.kr (J.E.P.); tamizhiniyan@hotmail.co.uk (T.I.G.); choy5181@gmail.com (Y.H.C.); leekho@chosun.ac.kr (K.H.L.); 2Department of Integrative Biological Sciences & BK21 FOUR Educational Research Group for Age-Associated Disorder Control Technology, Chosun University, Gwangju 61452, Korea; 3Gwangju Alzheimer’s Disease and Related Dementias Cohort Center, Chosun University, Gwangju 61452, Korea; khaser@chosun.ac.kr (K.Y.C.); jjjlee21@gmail.com (J.J.L.); 4Department of Neurology, Chonnam National University Medical School, Gwangju 61469, Korea; drchoism@gmail.com (S.-M.C.); k906141h@hanmail.net (S.H.C.); 5Department of Neurology, Chonnam National University Hospital, Gwangju 61469, Korea; smksarang@hanmail.net; 6Department of Nuclear Medicine, Chonnam National University Medical School and Hospital, Gwangju 61469, Korea; jahaekim.md@gmail.com (J.K.); songhc@jnu.ac.kr (H.-C.S.); 7Laboratory of Functional and Medicinal Proteomics, School of Life Sciences, Gwangju Institute of Science and Technology, Gwangju 61005, Korea; zeeyong@gist.ac.kr; 8Cell Logistics and Silver Health Research Center, School of Life Sciences, Gwangju Institute of Science and Technology, Gwangju 61005, Korea; wksong@gist.ac.kr; 9Department of Physiology, Chonnam National University Medical School, Hwasun 58128, Korea; jhsjeong@hanmail.net; 10Aging Neuroscience Research Group, Korea Brain Research Institute, Daegu 41062, Korea

**Keywords:** Alzheimer’s disease, plasma biomarkers, NFL, Aβ_1–42_, combinatorial biomarkers

## Abstract

Potential biomarkers for Alzheimer’s disease (AD) include amyloid β_1–42_ (Aβ_1–42_), t-Tau, p-Tau_181_, neurofilament light chain (NFL), and neuroimaging biomarkers. Their combined use is useful for diagnosing and monitoring the progress of AD. Therefore, further development of a combination of these biomarkers is essential. We investigated whether plasma NFL/Aβ_1–42_ can serve as a plasma-based primary screening biomarker reflecting brain neurodegeneration and amyloid pathology in AD for monitoring disease progression and early diagnosis. We measured the NFL and Aβ_1–42_ concentrations in the CSF and plasma samples and performed correlation analysis to evaluate the utility of these biomarkers in the early diagnosis and monitoring of AD spectrum disease progression. Pearson’s correlation analysis was used to analyse the associations between the fluid biomarkers and neuroimaging data. The study included 136 participants, classified into five groups: 28 cognitively normal individuals, 23 patients with preclinical AD, 22 amyloid-negative patients with amnestic mild cognitive impairment, 32 patients with prodromal AD, and 31 patients with AD dementia. With disease progression, the NFL concentrations increased and Aβ_1–42_ concentrations decreased. The plasma and CSF NFL/Aβ_1–42_ were strongly correlated (*r* = 0.558). Plasma NFL/Aβ_1–42_ was strongly correlated with hippocampal volume/intracranial volume (*r* = 0.409). In early AD, plasma NFL/Aβ_1–42_ was associated with higher diagnostic accuracy than the individual biomarkers. Moreover, in preclinical AD, plasma NFL/Aβ_1–42_ changed more rapidly than the CSF t-Tau or p-Tau_181_ concentrations. Our findings highlight the utility of plasma NFL/Aβ_1–42_ as a non-invasive plasma-based biomarker for early diagnosis and monitoring of AD spectrum disease progression.

## 1. Introduction

According to the International Population reports, the global elderly population accounted for 8.5% of the total population in 2018 and is predicted to nearly triple by 2050 [1]. The rapid aging of the world’s population has made dementia a significant research topic. Dementia associated with Alzheimer’s disease (AD) is underscored by a decline in cognitive function and accounts for >65% of all cases of dementia [2,3]. AD progression has become a major social issue as it affects the length and quality of the patients’ life and the psychological and economic status of patients and their families [4]. AD is a neurodegenerative disease characterized by abnormal accumulation of amyloid beta (Aβ) and tau protein in the brain. Currently available pharmacotherapies have been unable to halt the pathophysiological progression of AD. Therefore, early diagnosis and prevention of AD are crucial to reduce disease severity and improve prognosis [5]. Current tools for monitoring AD pathology include neuroimaging biomarkers using MRI or PET [6] and CSF biomarkers [6,7]. However, diagnostic and prognostic evaluation using neuroimaging biomarkers is expensive, and the availability of these biomarkers is limited [6]. Furthermore, evaluation using CSF biomarkers requires an invasive lumbar puncture and is associated with numerous other challenges related to monitoring disease progression and developing disease-modifying treatments [7]. Therefore, substantial research has been undertaken on various clinical biomarkers and combinations of biomarkers in recent years, and efforts are underway to develop non-invasive and quantitative approaches for measuring biomarkers.

The proteins Aβ_1–42_, t-Tau, and p-Tau_181_ have been employed as CSF biomarkers for AD [8]. In addition, neurofilament light chain (NFL) has recently attracted attention as a biomarker for neuroaxonal damage [9,10,11,12]. The ATN system (amyloid, tau, neurodegeneration) has been established as a multimodal classification scheme [13]. Plasma and CSF NFL concentration can be easily measured, highlighting the potential use of NFL as a candidate marker for tracking neurodegeneration in AD [12,14,15,16,17,18]. A longitudinal study reported that plasma NFL could, in fact, be used as a non-invasive biomarker to track neurodegeneration in patients with AD [19]. However, NFL concentration is known to increase in other neurodegenerative disorders, such as frontotemporal dementia, dementia with Lewy bodies, and corticobasal syndrome [20]. Therefore, there is an urgent need to identify disease-specific monitoring biomarkers for AD pathology and neurodegeneration.

In this study, we analysed CSF and plasma samples from patients in the five stages of the AD spectrum to identify potential biomarkers for AD. We measured NFL and Aβ_1–42_ as proxies of neurodegeneration and amyloid pathology, respectively. Further, we performed correlation analyses to examine the relationship between plasma and CSF NFL/Aβ_1–42_, and between plasma NFL/Aβ_1–42_ and brain atrophy. Further, we derived the cut-off values for plasma NFL/Aβ_1–42_ using receiver operating characteristic (ROC) curves to confirm its diagnostic ability. Here, we present plasma NFL/Aβ_1–42_ as a plasma-based primary screening biomarker reflecting brain neurodegeneration and amyloid pathology in AD that can be used for monitoring disease progression, early diagnosis, and in studies on early disease treatment.

## 2. Materials and Methods

### 2.1. Study Participants

Data were obtained from the Gwangju Alzheimer’s Disease and Related Dementias Cohort database in Gwangju, South Korea [21,22,23]. All research data and samples were collected between August 2015 and October 2017 in Gwangju and Jeollanam-do in Korea. All CSF and plasma samples used in this study were collected on the same day. The study was approved by the Chosun University Hospital Institutional Review Board (IRB approval numbers: 2013–12–018-068 and 2016–10–005-009). All study participants provided written informed consent, and the study procedures were performed following the ethical standards of the Helsinki Declaration. The clinical criteria of the National Institute of Neurological Disorders and Stroke/Diagnostic and Statistical Manual of Mental Disorders, version IV (NINDS/DSM-IV) were applied. We excluded only seven of the 143 patients who were initially screened for this study: two who were diagnosed with Lewy bodies, four with normal pressure hydrocephalus, and one with semantic variation of frontal temporal dementia. The final sample size for this study was 136 participants. The participants were classified into five groups: 28 cognitively normal individuals (CN Aβ−), 23 patients with preclinical AD (CN Aβ+), 22 amyloid-negative patients with amnestic mild cognitive impairment (aMCI Aβ−), 32 patients with prodromal AD (aMCI Aβ+), and 31 patients with AD dementia (AD Aβ+), according to the clinical criteria proposed by the International Working Group-2 (IWG-2) guidelines with amyloid-PET [24]. Inclusion and exclusion criteria are described in our previous reports [21,25,26].

### 2.2. MRI Acquisition and Processing

Brain MRI scans were acquired at the Chosun University Hospital and Chonnam National University Hospital. Among the 136 participants, one from the AD group and one from the prodromal AD group were excluded due to unavailability of MRI data. Detailed image acquisition protocols are described in our previous reports [27,28]. All MR T1-weighted images were processed with Freesurfer software version 5.3.0 (Laboratory for Computational Neuroimaging, Athinoula A. Martinos Center for Biomedical Imaging, Charlestown, MA, USA) (https://surfer.nmr.mgh.harvard.edu/fswiki, accessed on 13 December 2021) using an automated processing pipeline. Motion correction, normalization, non-brain tissue removal, white-matter (WM) and gray-matter (GM) segmentation, Talairach transformation, intensity normalization, topology correction, tessellation of GM and WM boundaries, and optimization of GM/WM and GM/CSF boundaries were performed in the Freesurfer automated processing software [29].

### 2.3. β- Amyloid PET Imaging and Processing

^18^F-Florbetaben (^18^F-FBB) PET amyloid imaging data were acquired from the Korean participants. Detailed image acquisition procedures are described in our previous reports [27,28]. Brain regions (frontal cortex, parietal cortex, temporal cortex, posterior cingulate/frontal cortex) were visually assessed and scored according to the brain beta-amyloid plaque load (BAPL) scoring system for PET scan. BAPL score sum of 1 is classified as beta-amyloid-negative, BAPL scores 2 and 3 are classified as beta-amyloid-positive PET scan [BAPL score: 1 = no beta-amyloid load, 2 = insignificant beta-amyloid load, 3 = significant beta-amyloid load] [30]. Among 136 participants, one patient from the AD group was excluded due to unavailability of PET data. All ^18^F-FBB PET images were processed using the SPM12 (https://www.fil.ion.ucl.ac.uk/spm/, accessed on 13 December 2021) toolbox installed in MATLAB (R2018a, Mathworks, Natick, MA, USA) software. ^18^F-FBB PET images were co-registered with T1-weighted MR images of the same participant that were acquired within 6 months or on the same day of acquisition of the amyloid PET images. Detailed processing steps are described in our previous report [27]. Standard uptake value ratio (SUVR) was calculated by quantifying cortical amyloid burden in six predefined cortical regions (lateral temporal, anterior and posterior cingulate, and frontal and lateral parietal) and normalized to amyloid burden in the whole cerebellum.

### 2.4. CSF ATN Biomarker Analysis

CSF collection and storage were performed as described previously [21,25,26]. CSF Aβ_1–42_, t-Tau, and p-Tau_181_ concentrations were quantified using an INNOTEST ELISA kit (Fujirebio, Ghent, Belgium), and NFL concentrations were measured using an NFL ELISA kit (UmanDiagnostics, Umea, Sweden), according to the protocols provided by the manufacturers.

### 2.5. Plasma NFL and Aβ_1–42_ Analysis

Plasma was collected from participants and stored according to the Molecular Medicine Ireland guidelines for standardized biobanking [29,31]. Plasma Aβ_1–42_ and NFL concentrations were measured at DNA Link (Seoul, Korea) using commercially available Aβ_1–42_ and NFL kits and the SiMoA HD-X analyser (all from Quanterix Inc., Billerca, MA, USA).

### 2.6. Statistical Analyses

Statistical analyses were performed using IBM SPSS version 26.0 (IBM Corp., Armonk, NY, USA) and GraphPad Prism version 5.00 (GraphPad Software Inc., La Jolla, CA, USA). Analysis of covariance was used to compare two (presence or absence of amyloid pathology) or three (CN, aMCI, and AD dementia) groups after adjusting for sex and age effects. Sex and APOE 4 carrier status (data missing for one case) were compared using a chi-square test for independent samples among the clinically defined groups. *p*-values < 0.05 were considered statistically significant. Cortical atrophy was evaluated using a general linear model implemented in the Surfstat toolbox (http://www.math.mcgill.ca/keith/surfstat/, accessed on 13 December 2021) in MATLAB (R2020b, The Mathworks, Natick, MA, USA). The effects of cortical atrophy in the CN Aβ+, aMCI Aβ+, and AD Aβ+ participants were compared with those in the CN Aβ- participants. Similarly, cortical atrophy in participants with MCI was compared with that in CN participants. A general linear model was used to assess cortical atrophy using age, sex, and field strengths as covariates. Pearson’s correlation analysis was used to analyse the associations between the fluid biomarkers and neuroimaging data. The ROC curves were generated using R studio (Boston, MA, USA). The optimum cut-off values from the ROC curves were determined using the Youden index. Dynamics of biomarkers were generated using SigmaPlot 10.0 (Systat Software Inc., Erkrath, Germany). The mean normalized biomarker concentrations were calculated as z-scores. 

### 2.7. Data Availability

Data not provided in the article because of space limitations may be shared (anonymized) at the request of any qualified investigator for purposes of replicating procedures and results.

## 3. Results

### 3.1. Fluid Biomarker Concentrations and Demographic Data

Fluid biomarker concentrations and neuroimaging data were analyzed for a total of 136 participants, comprising cognitively normal participants (CN) (n = 51), those with amnestic mild cognitive impairment (aMCI) (n = 54), and those with AD dementia (n = 31). Demographic data of the study participants are presented in Table 1 and Appendix A. CSF t-Tau and CSF p-Tau_181_ concentrations, CSF NFL/Aβ_1–42_, and plasma NFL/Aβ_1–42_ were significantly higher in the aMCI and AD dementia groups than in the CN group, whereas CSF Aβ_1–42_ and plasma Aβ_1–42_ concentrations were lower (*p* < 0.001, Figure 1a–h and Table 1). No significant differences were observed in the CSF and plasma NFL concentrations among the CN, aMCI, and AD dementia groups (Figure 1a,e). Significant differences in the CSF concentrations of the ATN biomarkers (CSF Aβ_1–42_, CSF t-Tau, CSF p-Tau_181_, and CSF NFL), plasma NFL, plasma Aβ_1–42_, and combination biomarkers (CSF NFL/Aβ_1–42_ and plasma NFL/Aβ_1–42_) were noted between all AD continuum groups and the CN Aβ- group (Figure 1a–h and Table 1).

### 3.2. Severe Brain Atrophy in Participants with AD Dementia 

The amyloid-PET SUVR scores increased with the progression of AD stage, except in the aMCI (Aβ−) group (Figure 1i). Hippocampal volume/intracranial volume (ICV) decreased progressively across the CN, aMCI, and AD dementia groups, and appeared to decrease with AD progression regardless of amyloid pathology (Figure 1j and Table 1). Cortical atrophy patterns in the CN Aβ+, aMCI Aβ+, and AD Aβ+ groups were compared with those in the CN Aβ− group (Figure 1k). No cortical atrophy was evident in the CN Aβ+ group. The aMCI Aβ+ group exhibited cortical atrophy in the precuneus region. Severe cortical atrophy was observed in the entorhinal cortex, precuneus, and lateral temporal lobe in patients with AD dementia (Figure 1k).

### 3.3. Association of Plasma NFL/Aβ_1–42_ with CSF NFL/Aβ_1–42_ and Hippocampal Volume/ICV in AD

CSF NFL concentrations were positively correlated with plasma NFL concentrations (*r* = 0.608, *p* < 0.001) (Figure 2a), which in turn were positively correlated with CSF t-Tau (*r* = 0.486, *p* < 0.001) and p-Tau_181_ concentrations (*r* = 0.502, *p* < 0.001) and negatively correlated with CSF Aβ_1–42_ concentrations (*r* = −0.259, *p* < 0.01) (Table 2). CSF Aβ_1–42_ concentrations were positively correlated with plasma Aβ_1–42_ concentrations (*r* = 0.472, *p* < 0.001) (Figure 2b) and negatively correlated with the amyloid-PET SUVR scores (*r* = −0.701, *p* < 0.001) (Table 2). Plasma NFL/Aβ_1–42_ was correlated with CSF ATN biomarker concentrations as a whole (*r* > 0.4), and strongly correlated with CSF NFL (*r* = 0.521, *p* < 0.001) and CSF Aβ_1–42_ (*r* = −0.462, *p* < 0.001) concentrations (Table 2). Plasma NFL/Aβ_1–42_ was strongly correlated with CSF NFL/Aβ_1–42_ (*r* = 0.562, *p* < 0.001) (Figure 2c and Table 2), and moderately correlated with hippocampal volume/ICV (*r* = −0.409, *p* < 0.001) and the Aβ-PET SUVR scores (*r* = −0.410, *p* < 0.001) (Figure 2d and Table 2).

### 3.4. Diagnostic Accuracy of Plasma NFL/Aβ_1–42_

No significant differences were observed in the CSF or plasma NFL concentrations among the CN, aMCI, and AD groups (Appendix A). However, a significant difference in the CSF and plasma NFL concentrations was noted in the diagnostic groups with amyloid pathology (Figure 1a,e, Table 1). CSF NFL concentrations were reflected in neuronal degeneration in the brains of patients with preclinical AD and were also rapidly reflected in the plasma (Figure 1e). In contrast, area under the curve (AUC) values that distinguished between the participants with AD (preclinical AD, AUC = 0.731; prodromal AD, AUC = 0.781; and AD dementia, AUC = 0.782) and the CN Aβ- group were similar. The AUC values for other CSF biomarkers (Aβ_1–42_, t-Tau, and p-Tau_181_) were significantly increased (Figure 2e–h and Table 3). The AUC values of plasma NFL concentrations (preclinical AD, AUC = 0.668; prodromal AD, AUC = 0.696; and AD dementia, AUC = 0.710) that distinguished the AD groups from the CN Aβ- group were not significantly increased, whereas those of plasma Aβ_1–42_ concentrations were significantly increased (Figure 2i–l and Table 3). CSF NFL concentrations reflected neurodegeneration in the brain and were increased in patients at the preclinical stage of AD (Figure 1a). These changes were rapidly reflected in the plasma (Figure 1e). In contrast, the AUC values that distinguished AD stages were similar (Figure 2e–l). Decrease in the CSF Aβ_1–42_ concentrations was reflected in the plasma (Figure 1b,f), and the AUC values according to AD stage were also significantly increased (Figure 2e–l and Table 2). However, for improved diagnostic accuracy, analyses were performed using NFL/Aβ_1–42_ (∆ delta ratio) based on the difference between the amount of increase in NFL and decrease in CSF and plasma Aβ_1–42_ concentrations in the patient group compared to cognitive normal group.
Δ Plasma NFL/Aβ_1-42_ = Increased NFL/Decreased Aβ_1-42_(1)

The AUC values were significantly higher for plasma NFL/Aβ_1–42_ combination biomarkers than for single plasma biomarkers (plasma NFL or plasma Aβ_1–42_) (Table 3). The AUC values distinguishing participants with preclinical AD from CN Aβ- participants were 0.668 for plasma NFL concentrations (cut-off value > 17.3) and 0.741 for plasma Aβ_1–42_ concentrations (cut-off value < 10.45), whereas the AUC value for plasma NFL/Aβ_1–42_ increased to 0.791 (cut-off value > 1.7). The AUC values distinguishing participants with prodromal AD from CN Aβ- participants were 0.696 for plasma NFL concentrations (cut-off value > 19.0) and 0.748 for plasma Aβ_1–42_ concentrations (cut-off value < 9.3), whereas the AUC value for plasma NFL/Aβ_1–42_ increased to 0.865 (cut-off value > 2.05). The AUC values distinguishing participants with prodromal AD from those with aMCI Aβ- in the aMCI group were 0.650 for plasma NFL concentrations (cut-off value > 18.8) and 0.769 for plasma Aβ_1–42_ concentrations (cut-off value < 10.45), whereas the AUC value for plasma NFL/Aβ_1–42_ was 0.822 (cut-off value > 1.77) (Figure 2e–l and Table 3). The AUC values for plasma NFL/Aβ_1–42_ were higher than those for neuroimaging data (hippocampal volume/ICV and entorhinal cortex thickness) (Figure 2i–l and Table 3).

### 3.5. Dynamics of Biomarkers and Neuroimaging in AD

Changes in the mean z-values of AD fluid biomarkers and neuroimaging data according to the stage of AD are presented in Figure 3a. With progression of AD, the z-scores of CSF and plasma Aβ_1–42_ concentrations, hippocampal volume/ICV, and entorhinal cortex thickness decreased, whereas those of CSF NFL, CSF t-Tau, CSF p-Tau_181_, and plasma NFL concentrations and CSF NFL/Aβ_1–42_, plasma NFL/Aβ_1–42_, and Aβ-PET SUVR score increased (Figure 3a). Differences in the z-scores among biomarkers and neuroimaging measurements were compared using the ∆z-score (Figure 3b). In the CN group (CN Aβ- vs. preclinical AD), the ∆z-scores varied in the order of CSF Aβ_1–42_ concentrations, Aβ-PET SUVR score, CSF NFL/Aβ_1–42_, CSF NFL concentrations, plasma NFL/Aβ_1–42_, plasma Aβ_1–42_ concentrations, CSF t-Tau concentrations, plasma NFL concentrations, CSF p-Tau_181_ concentrations, and hippocampal volume/ICV (Figure 3b). In particular, the ∆z-score for plasma NFL/Aβ_1–42_ was 0.87, suggesting more rapid changes than that in CSF t-Tau (∆z-score = 0.55) and p-Tau_181_ (∆z-score = 0.49) concentrations in patients with preclinical AD (Figure 3b).

## 4. Discussion

The main findings of this study are that (1) plasma NFL/Aβ_1–42_ was correlated with CSF NFL/Aβ_1–42_ and hippocampal volume/ICV, (2) plasma NFL/Aβ_1–42_ was associated with higher diagnostic accuracy in the early stage of AD, and (3) plasma NFL/Aβ_1–42_ changed more rapidly than CSF t-Tau and CSF p-Tau_181_ concentrations in the preclinical stage of AD. Together, these results suggest that plasma NFL/Aβ_1–42_ may be used as a highly accurate biomarker for the early diagnosis and monitoring of disease progression in AD.

The pathological processes underpinning AD involve the accumulation of Aβ_1–42_ in the brain decades prior to the onset of clinical symptoms, followed by a decrease in cortical metabolism [32,33,34]. The diagnosis and prognosis of AD are currently dependent on expensive imaging approaches and neurophysiological tests [6,35]. CSF biomarkers directly reflect the brain environment and have been investigated for use in disease diagnosis and prognosis [7,36]. However, obtaining biomarker samples is an invasive process. Further, imaging biomarkers are typically evaluated at the stage of MCI when clinical symptoms are already present, thereby missing the optimal window for early treatment and prevention of AD. Therefore, ensuring timely treatment and prevention of AD is essential. To this end, developing biomarkers capable of early diagnosis in the preclinical stage of AD is crucial.

Previous studies have mainly focused on the use of NFL as a biomarker neurodegeneration, correlating with cognitive impairment and brain neuropathology [10,11,14,16,37]. However, this study is the first to observe changes in NFL and Aβ_1–42_ in all stages of the AD spectrum and suggests a plasma-based biomarker that simultaneously reflects Aβ_1–42_ pathology and neurodegeneration in the brain. Here, we evaluated CSF and plasma samples to identify potential biomarkers for AD. Our study provides several notable findings. We observed that CSF NFL and CSF Aβ_1–42_ concentrations were correlated with plasma NFL and plasma Aβ_1–42_ concentrations, respectively. Further, plasma NFL/Aβ_1–42_ was correlated with the currently approved AD-CSF biomarkers (Aβ_1–42_, t-Tau, and p-Tau_181_) and neuroimaging biomarkers (Aβ-PET and MRI). We also identified that plasma NFL/Aβ_1–42_ distinguished participants with preclinical AD from CN participants and that the difference in the plasma NFL/Aβ_1–42_ z-scores was greater than that for CSF p-Tau_181_ and CSF t-Tau concentrations in the preclinical stage of AD. 

Our observed correlations of NFL and Aβ_1–42_ concentrations in the CSF with those in the plasma are consistent with previous findings [38,39], suggesting that plasma NFL and Aβ_1–42_ are derived from the CNS in patients with AD. Our observations of increased CSF and plasma NFL concentrations from preclinical AD to AD dementia are in accordance with data shown in previous reports [16]. Further, the average AUC value distinguishing AD diagnostic groups was ~0.7. Increased CSF NFL concentrations were rapidly reflected in the plasma at the preclinical AD stage, though it did not affect the diagnostic ability to distinguish AD stages. CSF and plasma Aβ_1–42_ concentrations were less strongly correlated than CSF and plasma NFL concentrations, whereas the diagnostic accuracy for distinguishing AD stages gradually improved. In this regard, combination biomarkers (plasma NFL/Aβ_1–42_) were associated with improved diagnostic accuracy compared to individual biomarkers (plasma NFL or plasma Aβ_1–42_ concentrations).

The beta coefficient for CSF NFL concentrations has been known similar to those for plasma NFL concentrations [12]. It is well-established that changes in the plasma NFL concentrations are associated with changes in global cognition, attention, and amyloid-PET findings [12]. Further, AD-susceptible brain atrophy in the hippocampus and entorhinal cortex has been reported [40,41,42,43]. Previous ATN studies on CSF have reported atrophy in the hippocampus, entorhinal cortex, and temporal regions in patients with MCI and AD dementia [19]. Here, we noted strong patterns of brain atrophy in similar regions, including the hippocampus, entorhinal cortex, temporal lobe, and precuneus regions in ATN triple-positive patients with AD dementia. Further, we observed subtle atrophy in the precuneus region of patients in the prodromal AD stage group. Previous studies have included Caucasian populations, whereas this study included participants from a homogeneous Korean population. Our results indicate that structural brain atrophy on MRI occurred only after patients entered the stage of MCI, whereas axonal neurodegeneration underpinned by tau pathology was already present at the preclinical AD stage (i.e., early stage of AD). Collectively, these results suggest that abnormalities in the CNS commence prior to the manifestation of clinical symptoms of AD.

The limitations of currently available MRI-based approaches are evident [6]. As such, the development of plasma-based biomarkers for early diagnosis and disease-monitoring provides a key solution for the treatment and prevention of AD. Plasma NFL/Aβ_1–42_ exhibited excellent performance for differentiating stages of the AD spectrum, especially early stages of AD (Table 3). Compared to single biomarkers, the combination biomarker plasma NFL/Aβ_1–42_ demonstrated higher performance accuracy, highlighting its utility as a candidate biomarker for the early diagnosis of AD. In addition, our results imply that plasma NFL/Aβ_1–42_ may be used as a preliminary screening tool to identify patients who require precision medical testing, such as additional PET, MRI, or CSF analysis. NFL and Aβ_1–42_ concentrations were detected using single molecular array (an ultra-sensitive immunoassay method), which enables the detection of very short fragments (~10 kDa) or peptides. Further, stable fragments are highly reproducible and reliable as plasma-based biomarkers for monitoring neurodegeneration and disease progression. Additionally, the relative dynamics of fluid and imaging biomarkers measured in this study are in accordance with previous results [38,44]. Although the combination biomarker plasma NFL/Aβ_1–42_ exhibited slower dynamics than CSF Aβ_1–42_ concentrations, differences were observed at early stages of AD when compared to those observed with CSF t-Tau and p-Tau_181_ concentrations.

One limitation of study is that the diagnosis group underwent amyloid-PET, which enables differential diagnoses for AD. The increase in NFL concentrations was analyzed in relation to amyloid pathology; however, further studies examining tau pathology are warranted. In addition, the diagnostic ability of plasma NFL/Aβ1-42 and its capacity to distinguish AD from other dementias should be verified using verification cohorts and longitudinal studies. Authors should discuss the results and how they can be interpreted from the perspective of previous studies and of the working hypotheses. The findings and their implications should be discussed in the broadest context possible. Future research directions may also be highlighted.

## 5. Conclusions

Our results suggest that plasma NFL/Aβ_1–42_ may be used as a non-invasive plasma-based biomarker for the early diagnosis and monitoring of neurodegeneration in AD. Plasma NFL/Aβ_1–42_ is a promising candidate tool to evaluate underlying neuropathology in patients with AD and may be leveraged in future therapeutic studies for disease treatment.

## Figures and Tables

**Figure 1 biomedicines-10-00169-f001:**
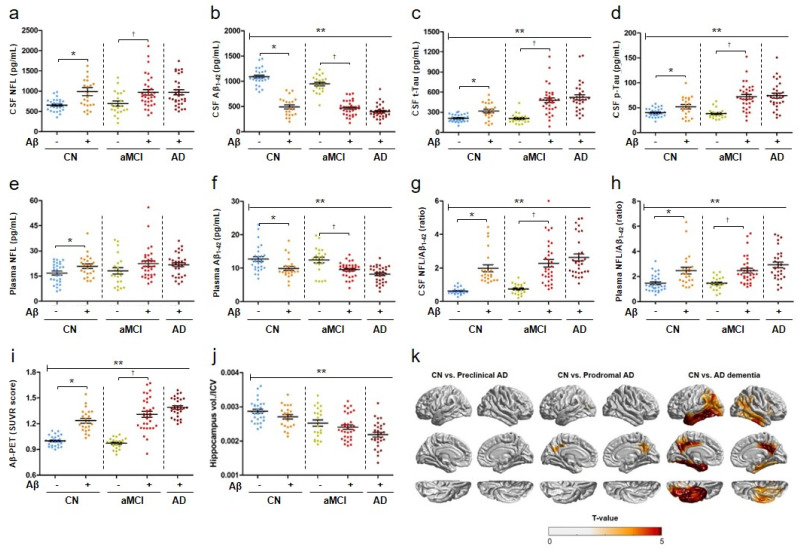
Biomarker concentrations in the CSF, plasma, and neuroimaging data. Data are presented as mean values of ATN (amyloid, tau, and neurodegeneration) biomarker concentrations in the CSF (**a**–**d**), plasma neurofilament light chain (NFL) concentrations (**e**), plasma Aβ_1–42_ concentrations (**f**), CSF NFL/Aβ_1–42_ (**g**), plasma NFL/Aβ_1–42_ (**h**), standard uptake value ratio (SUVR) scores (**i**), and value of hippocampal volume/intracranial volume (ICV) (**j**). Statistical analysis was performed using SPSS version 25. ** *p* < 0.001, statically significant group effect by ANOVA [groups: cognitively normal (CN) (n = 51), amnestic mild cognitive impairment (aMCI) (n = 54), and Alzheimer’s disease (AD) dementia (n = 31)]. * *p* < 0.005, ^†^
*p* < 0.05, significant difference between two indicated groups using ANCOVA adjusted for age and sex. (**k**) Brain cortical atrophy patterns as t-value maps in the preclinical AD, prodromal AD, and AD dementia groups. Preclinical AD (CN Aβ+) (n = 23), prodromal AD (aMCI Aβ+) (n = 32), and AD dementia (AD Aβ+) (n = 30) groups were compared with the CN Aβ− (n = 28) group to observe differences in point-wise cortical thickness using a general linear model with adjustments for age, sex, and field strength as covariates. Greater cortical atrophy was observed in the AD dementia group.

**Figure 2 biomedicines-10-00169-f002:**
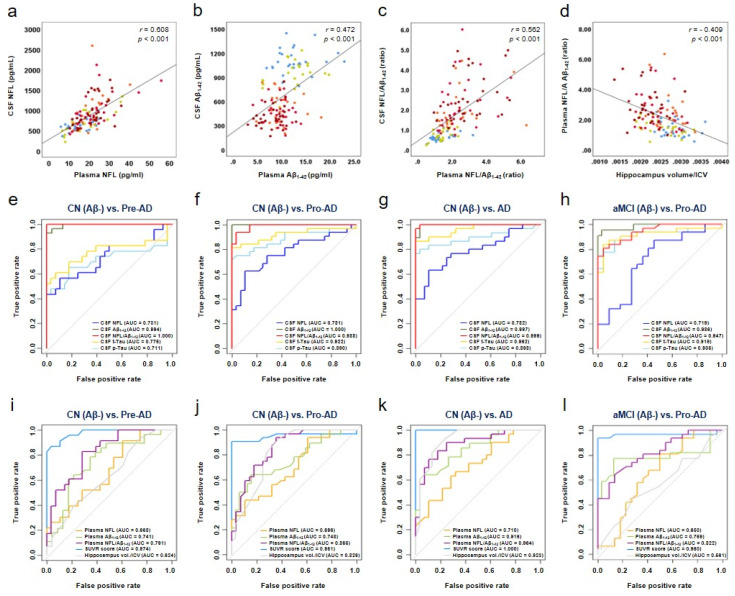
Correlation analysis, ROC curves, and biomarker dynamics. Pearson’s correlation analysis was used to analyse the correlations among CSF neurofilament light chain (NFL) and plasma NFL concentrations (**a**), CSF Aβ_1–42_ and plasma Aβ_1–42_ concentrations (**b**), CSF NFL/Aβ_1–42_ and plasma NFL/Aβ_1–42_ (**c**), and plasma NFL/Aβ_1–42_ and hippocampal volume/intracranial volume (ICV) (**d**). Representative ROC curves and AUC values are shown for indicated diagnostic groups (**e**–**l**). CSF and plasma biomarkers and neuroimaging dynamics as the standard uptake value ratio (SUVR) scores. Symbols: sky blue circle, CN(Aβ−); orange circle, Pre-AD; light green circle, aMCI(Aβ−); red circle, Pro-AD; dark red circle, AD.

**Figure 3 biomedicines-10-00169-f003:**
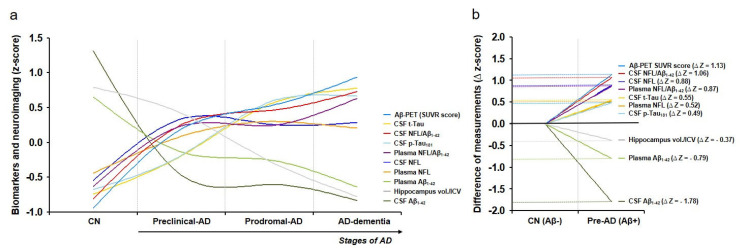
Dynamics of measurement. To compare biomarkers and neuroimaging data with different dynamic ranges, measurements were converted to z-scores (mean values of normalized biomarker levels of each group) based on the distribution in this study cohort. The plot indicates the mean z-scores for a given biomarker connected across progressively more affected diagnostic groups by a smoothing spin line using SigmaPlot 10.0 (**a**). The ∆z-score was calculated to compare the z-score differences between the cognitively normal (CN Aβ− and preclinical AD (CN Aβ+) groups (**b**).

**Table 1 biomedicines-10-00169-t001:** Patient demographics (n = 136).

Characteristics	Total	CN (n = 51)		aMCI (n = 54)		AD (n = 31)
		Aβ−	Aβ+ (Preclinical AD)	Aβ−	Aβ+ (Prodromal AD)	Aβ+
n	136	28	23	22	32	31
Age, mean (SD), y	136	69.4 (6.3)	73.9 (2.5) *	69.1 (8.6)	72.7 (8.0) *^†^	65.2 (8.7) *^†^
Education, mean (SD), y	132	9.3 (4.2)	11.2 (5.5)	10.4 (5.1)	9.6 (5.1)	6.4 (3.5) *^†^
Female sex, No. (%)	136	17 (60.7)	10 (43.5)	6 (27.3)	16 (50.0)	21.0 (67.7)
K-MMSE score, mean (SD)	131	26.5 (2.3)	27.5 (1.9)	25.7 (3.1)	24.8 (2.8) *	18.9 (4.3) *^†^
APOE ε4 carrier, No. (%)	133	3 (10.7)	16 (69.5) *	2 (9.1)	26 (81.3) *^†^	24 (77.4) *^†^
CSF biomarkers, mean (SD), pg/mL						
NFL concentrations, pg/mL	136	655.7 (150.0)	989.2 (487.5) *	693.7 (281.7)	960.5 (398.3) *^†^	970.4 (360.9) *^†^
Aβ_1–42_ con., pg/mL	136	1089.3 (160.2)	516.4 (192.3) *	947.3 (161.5)	473.0 (147.1) *^†^	399.1 (135.1) *^†^
t-Tau con., pg/mL	136	209.7 (54.6)	322.9 (122.1) *	206.8 (70.9)	475.7 (210.8) *^†^	522.8 (217.1) *^†^
p-Tau_181_ con., pg/mL	136	40.0 (8.8)	52.6 (19.4) *	38.2 (9.4)	72.2 (26.8) *^†^	74.4 (27.0) *^†^
Plasma biomarkers, mean (SD), pg/mL						
NFL con., pg/mL	136	16.7 (6.0)	20.9 (6.5) *	18.1 (9.1)	22.5 (9.3) *	21.8 (6.6) *
Aβ_1–42_ con., pg/mL	136	12.7 (3.9)	9.9 (3.0) *	12.4 (3.8)	9.5 (2.2) *^†^	8.2 (2.4) *^†^
Combination biomarkers, ratio						
CSF NFL/Aβ_1–42_ ratio	136	0.62 (0.17)	2.02 (1.0) *	0.74 (0.3)	2.29 (1.3) *^†^	2.62 (1.18) *^†^
Plasma NFL/Aβ_1–42_ ratio	136	1.46 (0.65)	2.46 (1.3) *	1.46 (0.5)	2.46 (1.1) *^†^	2.92 (1.19) *^†^
Neuroimaging						
Aβ- PET (SUVR score)	135	1.0 (0.06)	1.24 (0.13) *	0.97 (0.06)	1.30 (0.19) *^†^	1.3886 (0.11) *^†^
Hippocampal volume/ICV	134	0.0029 (0.00032)	0.0027 (0.00031)	0.0025 (0.00043)	0.0024 (0.00037) *	0.0021 (0.00038) *^†^
Entorhinal cortex (mm)	134	3.4213 (0.32244)	3.3355 (2.28296)	3.3066 (0.47337)	3.0681 (0.38707) *	2.9610 (0.45533) *^†^

Values are presented as means ± SD. Abbreviation: K-MMSE, Korean Mini-Mental State Examination; ICV, intracranial volume; SUVR, standardized uptake value ratio; CSF, cerebrospinal fluid; Aβ_1–42_, amyloid beta_1–42_; t-Tau, total Tau protein, p-Tau, phosphorylated Tau protein; NFL, neurofilament light chain; CN, cognitive normal; aMCI, amnestic mild cognitive impairment; AD, Alzheimer’s disease. Significant difference analysis using ANCOVA were adjusted sex and age. * Indicates a significant difference between the indicated group and the amyloid-negative CN group. ^†^ Indicates a significant difference between the amyloid-negative aMCI group.

**Table 2 biomedicines-10-00169-t002:** Correlations between CSF biomarkers, plasma biomarkers, and neuroimaging data.

Molecules	CSF Biomarkers				Plasma Biomarkers		Combination Biomarkers		Neuroimaging Data		
	NFL	Aβ_1–42_	t-Tau	p-Tau_181_	NFL	Aβ_1–42_	CSFNFL/Aβ_1–42_	PlasmaNFL/Aβ_1–42_	Aβ− PET (SUVR)	Hippocampal Volume/ICV	Entorhinal Thickness
CSF NFL concentrations	1	−0.259 **	0.486 *	0.502 *	0.608 *	−0.110	0.710 *	0.521 *	0.334 *	−0.359 **	−0.194 **
CSF Aβ_1–42_ concentrations		1	−0.410 *	−0.357 *	−0.242 **	0.472 *	−0.736 *	−0.462 *	−0.701 *	0.340 *	0.245 *
CSF t-Tau concentrations			1	0.923 *	0.265 **	−0.305 *	0.491 *	0.382 *	0.617 *	−0.427 *	−0.378 *
CSF p-Tau_181_ concentrations				1	0.280 **	−0.304*	0.476 *	0.364 *	0.555 *	−0.392 *	−0.334 *
Plasma NFL concentrations					1	0.169 **	0.493 *	0.612 *	0.218 **	−0.432 *	−0.221 **
Plasma Aβ_1–42_ concentrations						1	−0.321 *	−0.503 *	−0.374 *	0.086	0.031
CSF NFL/Aβ_1–42_ ratio							1	0.562 *	0.580 *	−0.379 *	−0.213 **
Plasma NFL/Aβ_1–42_ ratio								1	0.410 *	−0.409 *	−0.132
Aβ− PET (SUVR score)									1	−0.348 *	−0.307 *
Hippocampal volume/ICV										1	0.622 *
Entorhinal thickness											1

Data are presented as Pearson’s correlation coefficient (*r*). Bold values indicate significant associations (*, *p* <0.001; **, *p* <0.01). Abbreviations: Aβ, amyloid-beta protein; t-Tau, total Tau protein; p-Tau, phosphorylated Tau protein; NFL, neurofilament light chain; ICV, Intracranial volume; SUVR, standardized uptake value ratio; CSF, cerebrospinal fluid.

**Table 3 biomedicines-10-00169-t003:** Diagnostic accuracy and cut-off values for differentiating patient groups.

		CSF Biomarker (pg/mL)				Plasma Biomarker (pg/mL)		Combination (∆, Delta Ratio)		Neuroimaging Data		
		NFL	Aβ_1–42_	t-Tau	p-Tau^181^	NFL	Aβ_1–42_	CSF NFL/Aβ_1–42_	Plasma NFL/Aβ_1–42_	Aβ− PET (SUVR)	Hippocampal Volume/ICV	Entorhinal Cortex
CN (Aβ−) versus Pre-AD	Cutoff	>696.2	<817.3	>241.5	>43.6	>17.3	<10.45	>0.89	>1.7	>1.0695	<0.0028	<3.3995
	Sen (%)	65.2	96.4	76.2	66.7	69.6	67.9	100.0	69.6	91.3	57.1	57.1
	Spe (%)	60.7	95.2	67.9	64.3	50.0	69.6	96.4	66.7	82.1	56.5	56.5
	AUC (95%CI)	0.731 (0.59–0.88)	0.994 (0.98–1.00)	0.776 (0.63–0.92)	0.711 (0.55–0.87)	0.668 (0.52–0.82)	0.741 (0.60–0.88)	1.000 (1.00–1.00)	0.791 (0.67–0.91)	0.974 (0.94–1.0)	0.624 (0.47–0.78)	0.598 (0.44–0.76)
	*p* value	0.005	<0.001	0.003	0.010	0.041	0.003	<0.001	<0.001	<0.001	0.130	0.233
CN (Aβ−) versus Pro-AD	Cut-off	>735.7	<745.6	>276.9	>48.8	>19.0	<9.3	>0.94	>2.05	>1.1015	<0.0026	<3.2835
	Sen (%)	75.8	100.0	84.8	81.8	63.6	84.6	93.9	72.2	90.6	75.0	75.0
	Spe (%)	71.4	93.9	85.7	82.1	57.1	61.1	96.4	76.9	92.9	75.0	74.2
	AUC (95%CI)	0.781 (0.66–0.90)	1.000 (1.00–1.00)	0.922 (0.85–0.99)	0.890 (0.80–0.98)	0.696 (0.57–0.83)	0.748 (0.58–0.92)	0.988 (0.97–1.00)	0.865 (0.74–0.99)	0.951 (0.89–1.00)	0.826 (0.72–0.93)	0.793 (0.68–0.91)
	*p* value	<0.001	<0.001	<0.001	<0.001	0.009	0.02	<0.001	0.001	<0.001	<0.001	<0.001
CN (Aβ−) versus AD dementia	Cut-off	>752.4	<620.0	>284.7	>52.4	>20.9	<8.5	>1.26	>2.30	>1.2075	<0.0025	<3.2675
	Sen (%)	71.0	100.0	87.1	83.9	64.5	84.6	96.8	93.8	100.0	85.7	75.0
	Spe (%)	75.0	93.5	89.3	89.3	67.9	75.0	100.0	92.3	100.0	83.3	73.3
	AUC (95%CI)	0.782 (0.67–0.90)	0.997 (0.99–1.00)	0.962 (0.92–1.00)	0.898 (0.81–0.97)	0.710 (0.58–0.84)	0.916 (0.82–1.00)	0.999 (0.99–1.00)	0.964 (0.90–1.00)	1.00 (1.00–1.00)	0.923 (0.85–0.99)	0.804 (0.69–0.92)
	*p* value	<0.001	<0.001	<0.001	<0.001	0.006	<0.001	<0.001	<0.001	<0.001	<0.001	<0.001
aMCI (Aβ−) versus Pro-AD	Cut-off	>763.6	<745.6	>259.5	>45.0	>18.8	<10.45	>1.08	>1.77	>1.0545	<0.0024	<3.2530
	Sen (%)	66.7	95.5	87.9	84.8	63.6	77.3	84.8	75.0	90.6	54.5	68.2
	Spe (%)	68.2	93.9	86.4	86.4	63.6	68.8	86.4	72.7	90.9	53.1	67.7
	AUC (95%CI)	0.719 (0.58–0.86)	0.986 (0.96–1.00)	0.919 (0.84–0.99)	0.905 (0.82–0.99)	0.650 (0.49–0.81)	0.769 (0.62–0.92)	0.947 (0.90–0.99)	0.822 (0.71–0.93)	0.960 (0.90–1.00)	0.561 (0.40–0.72)	0.717 (0.56–0.87)
	*p* value	0.006	<0.001	<0.001	<0.001	0.061	0.001	<0.001	<0.001	<0.001	0.449	0.009

Statistically-derived optimal cut-off values were determined with the best balance between sensitivity (Sen) and specificity (Spe) values. Discrimination of prodromal AD and AD dementia groups from the cognitively normal group was performed using receiver operating characteristic (ROC) curve analysis and quantified by the area under the curve (AUC) using SPSS software version 24.0.

## Data Availability

For original data, please contact byeong.kim7@gmail.com. Detailed participant demographics may be found in “Appendix A” available with the online version of this article.

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
