# Peer review of "Diagnostic Blood Biomarkers in Alzheimer’s Disease"

_biomedicines, 2022, doi:10.3390/biomedicines10010169_

Round 1

Reviewer 1 Report

Park et al. have in this manuscript investigated the use of plasma and CSF biomarkers in persons along the AD spectrum, i.e. from Aβ-negative cognitive normal persons to patients diagnosed with AD dementia. Among the main findings brought forward is that plasma NFL/Aβ1-42 ratio could be used to predict AD progression in their cohort. The manuscript is clearly written and the findings are novel. The authors have made good use of biomarker and imaging data in their study and the manuscript presents the data in appropriate figures and tables.

Minor comments:

1. Materials and Methods: Please provide more information regarding how participants were stratified as being Aβ-positive or negative.

2. Results: Dhiman et al 2020 (Alzheimers Dement (Amst). 2020 Feb 27;12(1):e12005.) showed that CSF NFL and NFL/Aβ1-42 predicted cognition measured using MMSE. Cognitive data is also available for the current cohort, such as K-MMSE. The authors could potentially also have other cognitive test data of different cognitive domains (memory, speed/attention etc). Please perform statistical analyses of NFL and NFL/ Aβ in relation to cognitive data available, including K-MMSE.

3. Results: Underscore (_) is unnecessary when writing “plasma_NFL” and “CSF_NFL”. Please change this to “plasma NFL” and “CSF NFL” throughout the text and figures/tables and supplemental material.

4. Table 1: Change ADD to “AD dementia” as written in Suppl table 1. Also, this is the way AD is referred to throughout the text, and not as ADD. In the table legend, “dementia” is misspelled on line 229.

5. Table 1: To organize all tables similarly, please write pg/mL in parenthesis as done in Suppl table 1, and remove commas before (pg/mL) in Table 1 and Suppl table 1.

6. Figure 2: Panel (m) is referred to at the end of the legend but this is not included as a graph in this figure.

7. Discussion: Please expand the discussion of the findings in relation to other published papers on plasma and CSF NFL, both NFL alone as biomarker and in ratios with Aβ (for example discuss data in relation to findings in Dhiman et al, Alzheimers Dement (Amst). 2020 Feb 27;12(1):e12005.)

8. Please perform a thorough proof-reading of all parts of the manuscript, e.g. correct “werereflected” in section 3.4 line 252.

Author Response

  1. Materials and Methods: Please provide more information regarding how participants were stratified as being Aβ-positive or negative.

A: Thank you for your valuable comment. We have more described the stratification criteria for Aβ-positive or negative in Material and Methods section. “Brain regions (frontal cortex, parietal cortex, temporal cortex, posterior cingulate/frontal cortex) were visually assessed and scored according to the brain beta-amyloid plaque load (BAPL) scoring system for PET scan. BAPL score sum of 1 is classified as beta-amyloid-negative, BAPL scores 2 and 3 are classified as beta-amyloid-positive PET scan [BAPL score: 1 = no beta-amyloid load, 2 = insignificant beta-amyloid load, 3 = significant beta-amyloid load] [30].”

  1. Results: Dhiman et al 2020 (Alzheimers Dement (Amst). 2020 Feb 27;12(1):e12005.) showed that CSF NFL and NFL/Aβ1-42 predicted cognition measured using MMSE. Cognitive data is also available for the current cohort, such as K-MMSE. The authors could potentially also have other cognitive test data of different cognitive domains (memory, speed/attention etc). Please perform statistical analyses of NFL and NFL/Aβ in relation to cognitive data available, including K-MMSE.

A: We agree with you because your comments are so valuable and important. Therefore, we are going to analyze the correlation between NFL/Aβ1-42 and other cognitive test data of different cognitive domains (memory, speed/attention, etc) through further study.

  1. Results: Underscore (_) is unnecessary when writing “plasma_NFL” and “CSF_NFL”. Please change this to “plasma NFL” and “CSF NFL” throughout the text and figures/tables and supplemental material.

A: Thank you for your comment. We have removed unnecessary Underscore (_).

  1. Table 1: Change ADD to “AD dementia” as written in Suppl table 1. Also, this is the way AD is referred to throughout the text, and not as ADD. In the table legend, “dementia” is misspelled on line 229.

A: Thank you for your kind comment. We have changed in line 235.

  1. Table 1: To organize all tables similarly, please write pg/mL in parenthesis as done in Suppl table 1, and remove commas before (pg/mL) in Table 1 and Suppl table 1.

A: Thank you for your kind comment, we have unified the format of Table and Suppl table 1.

  1. Figure 2: Panel (m) is referred to at the end of the legend but this is not included as a graph in this figure.

A: Thank you for your kind comment. We have deleted (m).

  1. Discussion: Please expand the discussion of the findings in relation to other published papers on plasma and CSF NFL, both NFL alone as biomarker and in ratios with Aβ (for example discuss data in relation to findings in Dhiman et al, Alzheimers Dement (Amst). 2020 Feb 27;12(1):e12005.)

A: We have expended and added Ref [37] in the discussion section based on your very valuable comments.

  1. Please perform a thorough proof-reading of all parts of the manuscript, e.g. correct “werereflected” in section 3.4 line 252.

A: Thank you for your kind comment we have corrected the sentence in section 3.4 line 257.

Reviewer 2 Report

biomedicines-1528986: Diagnostic blood biomarkers in Alzheimer's disease

This report excellently showed that plasma NFL/Abeta1-42 is a good candidate of the blood marker for early diagnosis and monitoring of AD spectrum disease progression. The experiments were performed carefully and there are enough data to support the conclusion. Hence this report is worth being published in the Journal almost in the present form. Some minor revisions are suggested as follows, which may help to improve the value of this report.

Major Points
(1) The individual correlation between plasma NFL/Abeta1-42 and amyloid burden.
In Fig.1 the comparisons were performed among the groups (CN, aMCI and AD). The important point is that NFL/Abeta1-42 reflets the amyloid burden in each patient. For example, the graphs with Y axis of NFL/Abeta1-42 and X axis of amyloid burden (PET amyloid imaging) as Fig 2d may be informative with a correlation analysis.
(2) The relationship between NFL/Abeta1-42 and the clinical stage of dementia.
There is a big controversy that amyloid has little correlation with dementia severity, but the point is tau pathology. Similarly, in the present report, little difference was observed in NFL or Abeta1-42 concentration among CN (amyloid+), aMCI (amyloid+) and AD. To discuss this point, the correlation between NFL/Abeta1-42 and the severity of dementia in aMCI and AD should be examined. Otherwise, NFL/Abeta1-42 is only a maker of amyloid burden but not a maker of AD.
(3) The calculation of AUC and the measurement of NFL increase or Abeta1-42 decrease.
In the section "3.4. Diagnostic accuracy of plasma_NFL/Aβ1-42", AUC and NFL/Abeta1-42 change were discussed. The frequent blood and CSF sampling for a long time should be necessary in each patient to calculate AUC. This condition should be explained in the "2.1. Study participants" The graph of typical data with Y axis (NFL or Abeta1-42 concentration) and X axis (time) may be shown. In addition, the calculation method of NFL increase and Abeta1-42 decrease should be clearly described. For example, the decrease means the difference between the NFL concentration of heathy stage and that of AD stage in each patient.
(4) CN (amyloid+) aMCI (amyloid+) and AD
Little difference of NFL/Abeta1-42 seems to be observed among CN (amyloid+), aMCI (amyloid+) and AD in Fig 1g. From this results, NFL/Abeta1-42 can not distinguish CN and AD. Or is it reasonable that CN (amyloid+) will surely develop AD in near future? Is there such data in the cohort examined in the present report? The comparison of NFL/Abeta1-42 between AD and non-AD dementia may be added. NFL/Abeta1-42 should be useful to distinguish AD from other types of dementia. These matters should be discussed.

Minor Points
(a) It should be mentioned in "2.1. Study participants" whether the CSF sampling, the blood sampling, and imaging in each patient were performed in the same day or not. 
(b) The title should be more informative to show that the proposed marker is the blood NFL/Abeta1-42. The present title looks as those of some reviews. Hence, the title as follows can be suggested: "NFL/Abeta1-42 is the reliable diagnostic blood biomarker in Alzheimer's disease"
(c) In Fig.2 a, b, c, d, the meaning of different color circles should be indicated in the legend.
(d) Line 360, The <beta> estimates for CSF NFL concentrations: <beta> is hard to understand.
(e) Line 252, <werereflected> in neuronal degeneration: <werereflected> should be < were reflected>.

End of File

Author Response

(1) The individual correlation between plasma NFL/Abeta1-42 and amyloid burden. In Fig.1 the comparisons were performed among the groups (CN, aMCI and AD). The important point is that NFL/Abeta1-42 reflets the amyloid burden in each patient. For example, the graphs with Y axis of NFL/Abeta1-42 and X axis of amyloid burden (PET amyloid imaging) as Fig 2d may be informative with a correlation analysis. (2) The relationship between NFL/Abeta1-42 and the clinical stage of dementia. There is a big controversy that amyloid has little correlation with dementia severity, but the point is tau pathology. Similarly, in the present report, little difference was observed in NFL or Abeta1-42 concentration among CN (amyloid+), aMCI (amyloid+) and AD. To discuss this point, the correlation between NFL/Abeta1-42 and the severity of dementia in aMCI and AD should be examined. Otherwise, NFL/Abeta1-42 is only a maker of amyloid burden but not a maker of AD.

A: Thank you for your kind comment. As your point out, the concentrations of NFL and Abeta1-42 were little difference among CN (amyloid+), aMCI (amyloid+) and AD. Therefore, in this study, we focused on the biomarker NFL/Abeta1-42, which combines the characteristics of increasing NFL and decreasing Abeta1-42, to suggest a clear difference (p < 0.001) according to AD stage.

(3) The calculation of AUC and the measurement of NFL increase or Abeta1-42 decrease. In the section "3.4. Diagnostic accuracy of plasma_NFL/Aβ1-42", AUC and NFL/Abeta1-42 change were discussed. The frequent blood and CSF sampling for a long time should be necessary in each patient to calculate AUC. This condition should be explained in the "2.1. Study participants" The graph of typical data with Y axis (NFL or Abeta1-42 concentration) and X axis (time) may be shown. In addition, the calculation method of NFL increase and Abeta1-42 decrease should be clearly described. For example, the decrease means the difference between the NFL concentration of heathy stage and that of AD stage in each patient.

A: Thank you for your valuable comment. According to your suggestion, we have modified the sentence to make it clearer.

(4) CN (amyloid+) aMCI (amyloid+) and AD. Little difference of NFL/Abeta1-42 seems to be observed among CN (amyloid+), aMCI (amyloid+) and AD in Fig 1g. From this results, NFL/Abeta1-42 can not distinguish CN and AD. Or is it reasonable that CN (amyloid+) will surely develop AD in near future? Is there such data in the cohort examined in the present report? The comparison of NFL/Abeta1-42 between AD and non-AD dementia may be added. NFL/Abeta1-42 should be useful to distinguish AD from other types of dementia. These matters should be discussed.

A: Thank you for your valuable comment. As shown in Table 3, NFL/Abeta1-42 can distinguish CN and AD. The use of NFL/Abeta1-42 to differentiate Alzheimer’s disease from other types of dementia is very important. Therefore, we plan to do it through future study.

Minor Points :
(a) It should be mentioned in "2.1. Study participants" whether the CSF sampling, the blood sampling, and imaging in each patient were performed in the same day or not. 

A: We have added the sentence to make it clearer in section “2.1. Study participants”.

(b) The title should be more informative to show that the proposed marker is the blood NFL/Abeta1-42. The present title looks as those of some reviews. Hence, the title as follows can be suggested: "NFL/Abeta1-42 is the reliable diagnostic blood biomarker in Alzheimer's disease"

A: Thanks for your comments. We had a discussion, however we decided to keep the current title as it was appropriate. We ask for your understanding.

(c) In Fig.2 a, b, c, d, the meaning of different color circles should be indicated in the legend.

A: Thank you for your kind comment. We have added the information of color circles in Figure 2 legend.

(d) Line 360, The <beta> estimates for CSF NFL concentrations: <beta> is hard to understand.

A: According to your comment, we have changed the sentence.

(e) Line 252, <werereflected> in neuronal degeneration: <werereflected> should be < were reflected>.

A: Thank you for your kind comment we have corrected the sentence in section 3.4 line 257.